# Magnetization, Band Gap and Specific Heat of Pure and Ion Doped MnFe$_2$O$_4$ Nanoparticles

**I. N. Apostolova** [1], **A. T. Apostolov** [2] **and J. M. Wesselinowa** [3,*]

1   Faculty of Forest Industry, University of Forestry, Kl. Ohridsky Blvd. 10, 1756 Sofia, Bulgaria
2   Faculty of Hydrotechnics, University of Architecture, Civil Engineering and Geodesy,
    Hristo Smirnenski Blvd. 1, 1046 Sofia, Bulgaria
3   Faculty of Physics, University of Sofia, J. Bouchier Blvd. 5, 1164 Sofia, Bulgaria
*   Correspondence: julia@phys.uni-sofia.bg

**Abstract:** We have studied the magnetic properties of ion doped MnFe$_2$O$_4$ nanoparticles with the help of a modified Heisenberg model and Green's function theory taking into account all correlation functions. The magnetization $M_s$ and the Curie temperature $T_C$ increase with decreasing particle size. This is the opposite behavior than that observed in CoFe$_2$O$_4$ and CoCr$_2$O$_4$ nanoparticles. By Co, Mg or Ni doping, $M_s$ and $T_C$ increase with enhancing the dopant concentration, whereas, by La or Gd doping, the opposite effect is obtained due to the different doping and host ionic radii which change the exchange interaction constants. The band gap energy $E_g$ is calculated from the s–d model. It can decrease or increase by different ion doping. The peak observed in the temperature dependence of the specific heat at $T_C$ is field dependent.

**Keywords:** MnFe$_2$O$_4$ nanoparticles; magnetization; Curie temperature; band gap energy; specific heat; microscopic model

## 1. Introduction

Manganese ferrite MnFe$_2$O$_4$ (MFO) nanoparticles (NPs) attract the attention of researchers with remarkable soft-magnetic properties, good chemically stability and biocompatibility [1–8]. The last property is useful for application of MFO NPs in the magnetic hyperthermia for cancer therapy [9–12]. MFO crystallises in a normal spinel cubic structure with two sublattices—A and B. Mn$^{3+}$ and Fe$^{2+}$ prefer to occupy the octahedral sites while Mn$^{2+}$ the tetrahedral ones. MFO undergoes two magnetic transitions: one is the paramagnetic to ferrimagnetic transition at $T_C \sim 575$ K and the other is the spin-spiral transition temperature at low temperatures [13,14]. NPs have different properties compared to those of the bulk compound. For MFO, some controversial results are observed which are not clarified. For example, the reported behavior by many authors of the spontaneous magnetization $M_s$ and the phase transition temperature $T_C$ is different; they can increase [2] or decrease [6] with decreasing NP size. The properties of MFO can be modified by ion doping (for example, Co, Cr, Ni, Cu, La) at the Mn or Fe site [15–18], which can lead to different applications—for example, Co-doped MFO for energy storage applications as well as electrochemical supercapacitors [19]. Zhao et al. [20] studied the magnetic properties of strongly correlated transition metal oxides.

There are not so many theoretical papers about MFO. The spinel structure of bulk MFO is investigated using density functional theory (DFT) by Singh et al. [21]. Elfalaky and Soliman [22] studied the magnetic properties of bulk MFO using the generalized gradient approximation. The cation and magnetic orders in bulk MFO are observed from DFT by Huang et al. [23]. Rafiq et al. [24] investigated by first-principles approach the magnetic and optical properties of bulk MFe$_2$O$_4$ (M = Mn, Co, Ni) ferrites. Ab initio study of the magnetocrystalline anisotropy in bulk MFO is reported by Islam et al. [25]. Exchange

integrals and electronic structure of bulk MFO are calculated by Zuo et al. [3]. The magnetic properties of bulk MFO are studied using ab initio calculations by Mounkashi et al. [26].

In our previous work [27], the magnetization, polarization and band gap energy of $CuCr_2O_4$ (CCO) NPs are investigated. In the present paper, we will study for the first time the magnetization, band gap energy and specific heat of ion doped MFO—bulk and NPs using a microscopic model and the Green's function theory. We will clarify the observed discrepancies. The properties of MFO NPs are compared with those of $CoFe_2O_4$ (CFO) and CCO NPs. It should be noted that the most theoretical papers studied the bulk MFO compounds (and not the nanostructures) using the DFT which is mainly concerned with ground state properties at zero temperature, whereas we are able to make a finite temperature analysis of the excitation spectrum and of all physical quantities.

## 2. Model and Methods

The spinel ferrites crystallize in the face centered cubic spinel structure. In the normal spinel configuration, the $M^{2+}$ ions occupy the tetrahedral sites, while the octahedral sites contain the $Fe^{3+}$ ions [13]. The modified Heisenberg Hamiltonian $H_m$ describes the magnetic properties of ion doped MFO:

$$H_m = -\frac{1}{2}\sum_{i,j}(1-x)J_{ij}\mathbf{S}_i \cdot \mathbf{S}_j - \sum_{ij}xJ_{ij}^{Fe-DI}\mathbf{S}_i^{Fe} \cdot \mathbf{S}_j^{DI} - \sum_i D_i(S_i^z)^2 - g\mu_B\sum_i \mathbf{h} \cdot \mathbf{S}_i, \quad (1)$$

where $\mathbf{S}_i$ is the Heisenberg operator of the Fe ion at the lattice site $i$. The spin interaction $J$ between the two sublattices A and B determines the ferrimagnetic order in MFO where $|J_{A-B}| > |J_{B-B}| \gg |J_{A-A}|$. $D$ is the single-ion anisotropy, $\mathbf{h}$ is an external magnetic field, and $x$ is the ion doping concentration.

Magnetically, the spinel ferrites display ferrimagnetic ordering where the total magnetization is observed from $M = (M^A + M^B)$. From the spin Green's function $\ll S_i^{+A,B}; S_j^{-A,B} \gg$, the sublattice magnetization $M^{A,B}$ for arbitrary spin value $S^{A,B}$ are calculated as follows:

$$\begin{aligned}
M^{A,B} &= \langle S^{zA,B} \rangle = \frac{1}{N^2}\sum_{i,j}\left[(S^{A,B}+0.5)\coth[(S^{A,B}+0.5)\beta E_{ij}^{A,B}]\right. \\
&\quad - \left. 0.5\coth(0.5\beta E_{ij}^{A,B})\right],
\end{aligned} \quad (2)$$

$\beta = 1/k_BT$. $E_{ij}^{A,B}$ is the excitation energy observed from the poles of the Green's functions, for example for the A subsystem:

$$\begin{aligned}
E_{ij} &= \left(\frac{2}{N'}\sum_m J_{im}(\langle S_m^- S_i^+ \rangle + 2\langle S_m^z S_i^z \rangle)\delta_{ij} - 2J_{ij}(\langle S_i^- S_j^+ \rangle + 2\langle S_i^z S_j^z \rangle)\right. \\
&\quad + \left. 2D_i(2\langle S_i^z S_i^z \rangle - \langle S_i^- S_i^+ \rangle)\delta_{ij} + 2g\mu_B H\langle S_i^z \rangle\delta_{ij}\right)/2\langle S_i^z \rangle\delta_{ij},
\end{aligned} \quad (3)$$

where $N'$ is the number of lattice sites.

For the approximate calculation of the Green's functions, we use a method proposed by Tserkovnikov [28]. It goes beyond the random phase approximation, taking into account all correlation functions. Moreover, this method allows us to calculate also the imaginary part of the Green's function. We want now sketch it briefly. After a formal integration of the equation of motion for the Green's function

$$G_{ij}(t) = \langle\langle a_i(t); a_j^+ \rangle\rangle \quad (4)$$

one obtains

$$G_{ij}(t) = -i\theta(t)\langle[a_i; a_j^+]\rangle\exp(-i\omega_{ij}(t)t), \quad (5)$$

with

$$\omega_{ij}(t) = \omega_{ij} \quad - \quad \frac{i}{t}\int_0^t dt't'\left(\frac{\langle[j_i(t);j_j^+(t')]\rangle}{\langle[a_i(t);a_j^+(t')]\rangle}\right.$$
$$\left. - \quad \frac{\langle[j_i(t);a_j^+(t')]\rangle\langle[a_i(t);j_j^+(t')]\rangle}{\langle[a_i(t);a_j^+(t')]\rangle^2}\right) \tag{6}$$

and $j_i(t) = \langle[a_i, H_{interaction}]\rangle$. The time-independent term

$$\omega_{ij} = \frac{\langle[[a_i, H];a_j^+]\rangle}{\langle[a_i;a_j^+]\rangle} \tag{7}$$

is the excitation energy in the generalized Hartree–Fock approximation. The time-dependent term in Equation (6) includes damping effects.

For the calculation of the band gap energy $E_g$ of MFO, we use the s–d model where to $H_m$ are added the following terms $H_{el}$ and $H_{m-el}$:

$$H_{el} = \sum_{ij\sigma} t_{ij}c_{i\sigma}^+c_{j\sigma}, \tag{8}$$

$t_{ij}$ is the hopping integral, $c_{i\sigma}^+$ and $c_{i\sigma}$ are Fermi-creation and -annihilation operators, as well as

$$H_{m-el} = \sum_i I_i \mathbf{S}_i \mathbf{s}_i, \tag{9}$$

$I$ is the s–d interaction. $\mathbf{s}_i$ is the spin operator of the conduction electrons at site $i$ and can be expressed as $s_i^+ = c_{i+}^+c_{i-}$, $s_i^z = (c_{i+}^+c_{i+} - c_{i-}^+c_{i-})/2$.

The band gap energy $E_g$ is the energy difference between the valence and conduction bands:

$$E_g = \omega^+(\mathbf{k}=0) - \omega^-(\mathbf{k}=\mathbf{k}_\sigma). \tag{10}$$

$\omega^\pm(k)$ are the electronic energies

$$\omega^\pm(k) = \epsilon_k - \frac{\sigma}{2}IM, \tag{11}$$

$\sigma = \pm 1$. $\epsilon_k$ is the conduction band energy in the paramagnetic state.

## 3. Numerical Results and Discussion

The NP has a cubo-octahedral shape where a certain spin is fixed in the center of the NP, and all spins are ordered into shells numbered by $n = 1, \ldots, N$. The following model parameters are used for MFO which have a cubic spinel structure [15]: $J$(Mn-Mn) = 8.6 K, $J$(Fe-Fe) = 13.7 K, $J$(Mn-Fe) = -21.8 K [26], $D = -0.1$ meV, $T_C = 575$ K, $S = 5/2$, $I = 0.2$ eV, $t = 0.05$ eV. $J_{ij} = J(r_i - r_j)$ depends on the inverse proportional on the lattice parameters. The exchange interaction constant on the surface $J_s$ is different from that in the bulk $J_b$ due to the reduced symmetry and the changed number of next neighbors on the surface. Let us emphasize that the single-ion anisotropy in a small NP of MFO is observed to be nearly 20 times larger than the bulk value [29].

### 3.1. Size Dependence of the Magnetization and Curie Temperature

Figure 1 shows the temperature and size dependence of the spontaneous magnetization $M_s$. Upon cooling, for a bulk MFO, $M_s$ increases strongly below the Curie temperature $T_C = 575$ K, which is connected with the appearance of a collinear ferrimagnetic phase (curve 1). Below $T_C$ in $M_s(T)$, there is an anomaly around the spin-spiral transition temperature $T_S \sim 15$ K due to frustration which leads to a structural phase transition. Such

behavior is observed in the most ferrimagnetic spinels due to a transition by decreasing the temperature from a collinear to a noncollinear spin configuration [8,13,26].

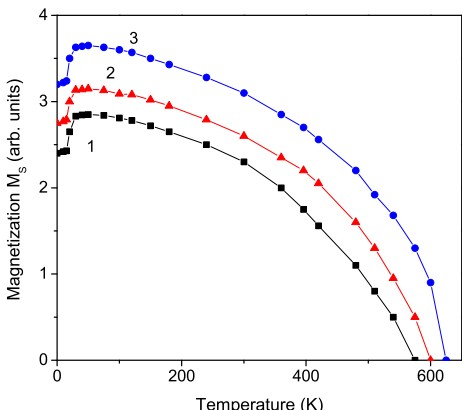

**Figure 1.** Temperature dependence of the magnetization $M_s$ in MFO for (1) bulk; and a NP with (2) $N = 20$ and (3) $N = 10$ shells, $J_s = 1.2\, J_b$, $D_s = 20\, D_b$.

In MFO, the spontaneous magnetization $M_s$ (Figure 1, curves 2 and 3) and the Curie temperature $T_C$ (Figure 2) increase with decreasing particle size due to finite size effects, to the breaking of Fe-O-Fe paths, whereas the spin-spiral transition temperature $T_S$ is nearly size independent (see Figure 1, curves 2 and 3). Furthermore, there appears an oxidation of $Mn^{2+}$ to $Mn^{3+}$ by the transition from the bulk state to nanoparticles [30]. The ionic radius of $Mn^{3+}$ (0.66 $\dot{A}$) is smaller than that of $Mn^{2+}$ (0.97 $\dot{A}$), which leads to smaller lattice parameters, i.e., $J_s > J_b$. The observed larger values of $M_s$ and $T_C$ for MFO NPs compared to the bulk ones are in good agreement with many authors [2,4,30–35]. In Figure 2, we have added some experimental data of Zheng et al. [34]. The authors have measured the magnetization versus temperature for MFO NPs and observed in the NP a 160 K higher Curie temperature than that in the bulk material. It can be seen that there is a good quantitative agreement between their experimental data [34] and our theoretical results. This is an indirect evidence for the appropriate chosen model and method. Let us emphasize that our results for $M_s(N)$ and $T_C(N)$ are in disagreement with those of Refs. [5,6,36,37]. It must be noted that, contrary to MFO, in CCO NPs [38–42] and in CFO NPs [4,39,43,44], a decrease of the spontaneous magnetization $M_s$ and the Curie temperature $T_C$ with decreasing NP size are observed.

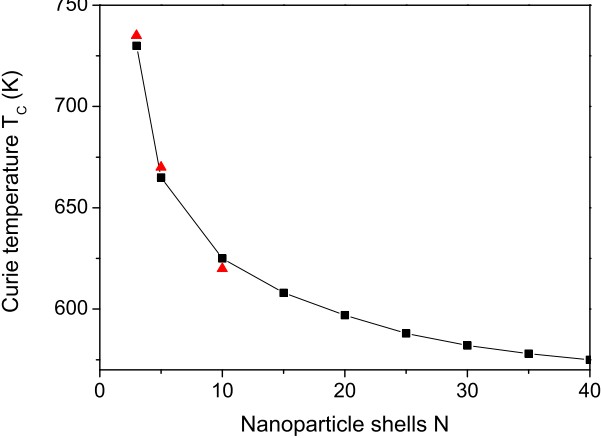

**Figure 2.** Size dependence of the Curie temperature $T_C$ with $J_s = 1.2\, J_b$, $D_s = 20\, D_b$. The red triangles are the experimental data from Ref. [34].



### 3.2. Ion Doping Dependence of the Magnetization and Curie Temperature

Let us study now the spontaneous magnetization $M_s$ and Curie temperature $T_C$ in $Co^{2+}$ ion doped MFO, $Mn_{1-x}Co_xFe_2O_4$ ($x = 0$–$0.5$), where the $Co^{2+}$ ions substitute the $Mn^{2+}$ ions on the octahedral sites. The radius of the $Co^{2+}$ ion (0.79 $\mathring{A}$) is smaller than that of the $Mn^{2+}$ ion (0.97 $\mathring{A}$), which leads to a compressive strain. Aslibeiki et al. [15] and Ansari et al. [45] have also obtained in the $Co^{2+}$ doped MFO NPs smaller lattice parameters (about 8.34 $\mathring{A}$) in comparison to the bulk one (8.51 $\mathring{A}$). Furthermore, there appears an oxidation of $Mn^{2+}$ to $Mn^{3+}$ by the transition from bulk to NPs, contributing to an effective cation distribution [30]. All this causes changes of the exchange interactions in the doped states (denoted with the index $d$ in our model). In the case of Co ion doping, we have the following relation $J_d > J_b$. Moreover, the magnetic anisotropy $D$ increases also strongly by the doping of Mn ions with Co ones. We observe that $M_s(x)$ and $T_C(x)$ (see Figure 3, curve 1) increase with the increasing of the Co ion doping concentration $x$. Let us emphasize that the larger spin moment of $Mn^{2+}$ ions ($S = 5/2$) compared to that of $Co^{3+}$ ($S = 3/2$), substituting $Co^{2+}$ on A sites could also lead to an increasing of the magnetization and affect the magnetic properties of MFO. This enhancement is in agreement with Aslibeiki et al. [15] but in disagreement with Reddy et al. [16], which reported enhanced lattice parameters and reduced magnetization with increasing Co substitution.

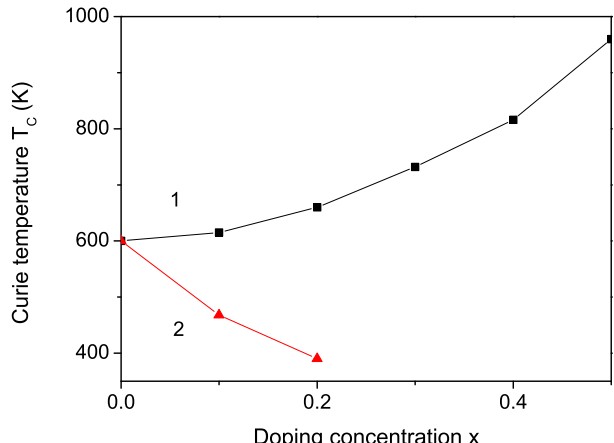

**Figure 3.** Dependence of the Curie temperature $T_C$ in CCO NP with $N = 20$ shells on the ion doping concentration $x$ for (1) Co with $J_d = 1.2J_b$; (2) La with $J_d = 0.8J_b$.

The magnetic properties of Mg ion doped MFO NPs are also studied. The ionic radius of $Mg^{2+}$ (0.72 $\mathring{A}$) is smaller compared to that of the $Mn^{2+}$ (0.80 $\mathring{A}$) ion. The lattice parameters decrease with increasing $x$, i.e., we have $J_d > J_b$, and observe an increase of $M_s$ in coincidence with the experimental data of Mg doped MFO NPs reported by Kombaiah et al. [46]. By $Ni^{2+}$ ion doping of MFO NPs, where the Ni ion radius (0.74 $\mathring{A}$) is again smaller compared to that of $Mn^{2+}$, Mathubata et al. [47] reported a decrease of the lattice parameters as well as of the spontaneous magnetization $M_s$. We would observe in Ni doped MFO a decrease of the lattice parameters but an increase of the magnetization $M_s$.

We investigate now the rare earth (RE) ion doping effect, for example $La^{3+}$, on the magnetization $M_s$ and Curie temperature $T_C$ in MFO, $MnFe_{2-x}La_xO_4$ ($x = 0$–$0.2$), where the La ions substitute only the Fe ions. The ionic radius of $La^{3+}$ (1.61 $\mathring{A}$) is larger than that of $Fe^{3+}$ (0.64 $\mathring{A}$), i.e., by La-doping, a tensile strain appears, and we have to choose $J_d < J_b$. This leads to a decrease of the spontaneous magnetization $M_s$ and Curie temperature $T_C$ (Figure 3, curve 2) with increasing La dopants. The decrease of the critical temperature $T_C$ could be useful for application in the magnetic hyperthermia, for cancer therapy, where $T_C$ must be about 310–315 K [48]. The magnetic behaviour of the ferrimagnetic oxides is mainly due to the 3d spin coupling in the Fe–Fe interaction. By introducing RE ions, a 3d–4f coupling in the RE–Fe interactions also appears, which changes the magnetic properties.

The RE–RE interactions are very weak and could be neglected. It must be mentioned that the single ion anisotropy of RE which slightly increases [15] by the RE ion doping of MFO is small in comparison with the cation rearrangement in MFO [49]. The reduction in the magnetic properties could be due to the additive magnetically weak $Fe^{+3}$-O-$Mn^{+2}$ interaction weakening the A–B interaction. We could also explain the decrease of the magnetic properties in a $Eu^{3+}$ or $Gd^{3+}$ doped MFO NP due to the larger radius of the $Eu^{3+}$ (0.95 $\dot{A}$) or $Gd^{3+}$ (0.94 $\dot{A}$) ions compared to that of $Fe^{3+}$, i.e., we obtain again a decrease of $M_s$ and $T_C$ with increasing $Eu^{3+}$ or $Gd^{3+}$ concentration. Moreover, by replacing $Fe^{3+}$ by paramagnetic $Eu^{3+}$ ions, the ferromagnetic region or super-exchange strength decreases. The observed decrease in $M_s$ and $T_C$ is in coincidence with the behaviour reported by La [50], Eu [51], Gd [52] doped MFO NPs, Tb doped $NiFe_2O_4$ [53], and Ce and Dy doped CFO [54], but in disagreement with the results in $Ho^{3+}$ substituted MFO NPs [55].

### 3.3. Ion Doping Dependence of the Band Gap Energy

Now, we study the ion doping effects on the optical band gap in MFO. Firstly, we will note that, due to the decreasing of the lattice parameters with decreasing NP size, i.e., we have $J_s > J_b$, the band gap energy $E_g$ decreases with decreasing size. Figure 4 presents the observed results by ion doping. The band gap energy $E_g$ of a MFO NP gradually decreases with an increase of Co doping (curve 1 for $J_d > J_b$) in agreement with Ansari et al. [45]. The origin of this decrease is that the ionic radius of $Co^{2+}$ (0.74 $\dot{A}$) is smaller than that of $Mn^{2+}$ (0.83 $\dot{A}$), i.e., the lattice parameters decrease in Co doped MFO NPs. The red triangles in Figure 4, curve 1, present the experimental data from Ref. [45]. The authors have determined the band gap energy from the UV-visible spectra. It can be seen that there is a good agreement with the experimental data of [45]. We would also obtain a similar decrease of the band gap energy $E_g$ in Mg or Ni doped MFO NPs, where the lattice parameters decrease with increasing the Mg or Ni content. This leads to a decrease of $E_g$ in coincidence with Kombaiah et al. [46] for Mg doped MFO NPs. However, Mathubata et al. [47] observed an increase of $E_g$ in Ni doped MFO NPs.

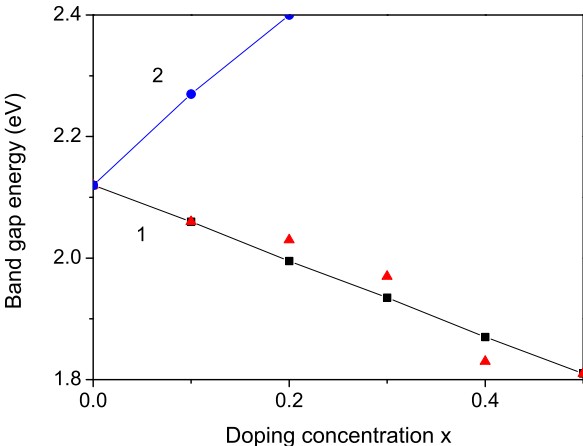

**Figure 4.** Dependence of the band gap energy $E_g$ of a MFO NP with $N$ = 20 shells on different ion doping: (1) Co; (2) La. The red triangles present the experimental data from Ref. [45].

On the other hand, by $La^{3+}$ ion doping of MFO with $J_d < J_b$, due to the larger radius of the $La^{3+}$ (1.61 $\dot{A}$) ion in comparison with that of $Fe^{3+}$ (0.64 $\dot{A}$), i.e., the lattice parameters increase, we obtain an increase of $E_g$ (curve 2) in coincidence with Kour et al. [56]. We would also observe an increase of $E_g$ by $Gd^{3+}$ ion doping of MFO NP due to the larger radius of the $Gd^{3+}$ (0.94 $\dot{A}$) ion compared to that of $Fe^{3+}$, i.e., there is again an enhancement of $E_g$ with increasing $Gd^{3+}$ dopants. The substitution of La and Gd improved the photocatalytic efficiency of nanoferrite MFO. Our result is in agreement with the observed enhancement of $E_g$ in Gd doped $NiFe_2O_4$ thin films [57] and Dy doped MFO NPs [58] but in disagreement with the observed decrease of $E_g$ in $La^{3+}$ and $Mo^{5+}$ doped MFO NPs [17,59].

### 3.4. Temperature and Magnetic Field Dependence of the Specific Heat

Finally, the specific heat in dependence on the temperature $C_p(T)$ of a MFO NP with $N = 10$ shells is studied. $C_p$ is observed from the equation $C_p = d\langle H \rangle / dT$, where $H$ is the full Hamiltonian. Longitudinal and transverse correlation functions appear which are calculated via the Spectral theorem from the corresponding Green's functions. Figure 5, curve 1, presents the result for $h = 0$. At the Curie temperature $T_C \sim 575$ K, where the $Fe^{3+}$ ion spins undergo a long-range ferrimagnetic ordering, there is a lambda-shaped peak in the specific heat. $C_p$ decreases with increasing magnetic field $h$ (see Figure 5, curves 1–3). The transition temperature $T_C$ decreases also with increasing magnetic field $h$ (curve 2). Applying a strong external magnetic field $h$, the peak disappears, and the transition is no longer observable (curve 3). The decrease of the peak at $T_C$ in the specific heat $C_p$ could be due to the depression of spin fluctuation enhancement by increasing the magnetic fields. A similar field dependent transition temperature in $C_p$ of MFO is reported by Balaji et al. [60].

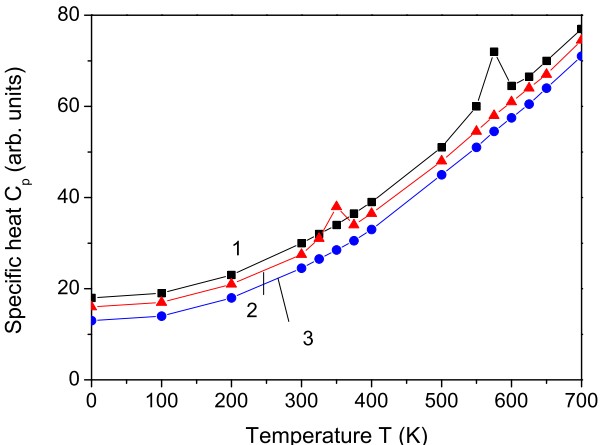

**Figure 5.** Temperature dependence of the specific heat $C_p$ of a MFO NP with $N = 10$ shells for different magnetic field $h$ values: 0 (1); 20 (2); 60 (3) kOe.

## 4. Conclusions

In conclusion, using the modified Heisenberg and the s–d models and the Green's function theory pure and ion doped MFO, bulk and NPs are investigated for the first time. MFO NPs are appropriate for various applications such as hyperthermia for cancer therapy, drug delivery, magnetic resonance imaging and storage devices. From the spontaneous magnetization $M_s$ in bulk MFO, as a function of the temperature, a ferrimagnetic transition at $T_C = 575$ K is observed, where $M_s$ vanishes, and an anomaly at low temperatures around the spin-spiral transition temperature $T_S \sim 15$ K corresponding to the helicoidal order temperature. The different strains appearing at the surface due to the reduced symmetry and the changed number of next neighbors as well as at the doped states due to the differences in the doping and host ionic radii lead to changes of the exchange interaction constants and therefore to changes of the properties in doped MFO NPs. Thus, we can consider the macroscopic physical quantities on a microscopic level. With decreasing particle size, the spontaneous magnetization $M_s$ and the Curie temperature $T_C$ decrease, whereas the spin-spiral transition temperature $T_S$ is nearly size independent. Due to a compressive strain by the substitution of the Mn ions on the octahedral sites with Co, Mg or Ni, the spontaneous magnetization $M_s$ and the Curie temperature $T_C$ increase with enhancing the doping concentration $x$, whereas, due to a tensile strain by the substitution of the Fe ions on the tetrahedral sites with La or Gd ions, we obtain the opposite effect. The decrease of the ferrimagnetic transition temperature $T_C$ could be useful for application in the magnetic hyperthermia, for cancer therapy, where $T_C$ must be about 310–315 K. The band gap energy $E_g$ decreases by Co, Mg or Ni ion substitution, but it increases by La or Gd ion doping. The substitution of La and Gd which enhances $E_g$ can be used for improvement

of the photocatalytic efficiency of nanoferrite MFO. The lower band gap by Co, Mg or Ni ion doping could increase the conductivity of the NP, which in turn can enhance its capacitance. A lambda-shaped peak appears in the specific heat at the critical temperature $T_C$, which vanishes applying high external magnetic fields.

Finally, we have tried to clarify some discrepancies in the reported experimental data. In our opinion, this could be due to the different synthesis and growth methodology, to the different doping methods and to the way of annealing.

As already mentioned in the Introduction due to their nontoxicity and biocompatiblity, MFO NPs are useful for application in the magnetic hyperthermia for cancer treatment [9–12]. In this context, the magnetic properties of MFO NPs for magnetic hyperthermia will be investigated in a future paper.

**Author Contributions:** All authors contributed equally to this work. All authors have read and agreed to the published version of the manuscript.

**Funding:** This research was funded by the Center for Research and Design of the University of Architecture, Civil Engineering and Geodesy (contract number BN-271/23).

**Institutional Review Board Statement:** Not applicable.

**Informed Consent Statement:** Not applicable.

**Data Availability Statement:** Derived data supporting the findings of this study are available from the corresponding author upon reasonable request.

**Acknowledgments:** A.T.A. acknowledges financial support from the Center for Research and Design of the Sofia University of Architecture, Civil Engineering and Geodesy (contract number BN-271/23).

**Conflicts of Interest:** The authors declare no conflict of interest.

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
