# Peer review of "Magnetization, Band Gap and Specific Heat of Pure and Ion Doped MnFe2O4 Nanoparticles"

_magnetochemistry, doi:10.3390/magnetochemistry9030076_

Round 1

Reviewer 1 Report

1. Conclusion can be improved by including all important findings and potential applications

2 In result and discussion certain result are found disagreeing with reported one. Its better to include reasons for the disagreement .

3.Experimental data is shown in graphs  but the details of experimental is not explained . it should be included in the manuscript

Author Response

Reviewer 1:

  1. “Conclusion can be improved by including all important findings and potential applications.”

The Conclusion is improved including all important results and potential applications of MFO NPs.

2 “In result and discussion certain result are found disagreeing with reported one. Its better to include reasons for the disagreement.”

At the end of the Conclusions we have included some possible reasons for the disagreement between some experimental data. In our opinion, although we are not experimentalists, this could be due to the experimental methodology of synthesis, of growth, to the method of doping, to the way of annealing.

3.”Experimental data is shown in graphs  but the details of experimental is not explained. It should be included in the manuscript.                  

We have included some details of the experimental data of the authors which are added in Figs. 2 and 4, see pp. 7 and 9. It can be seen that there is a good quantitative agreement between our theoretical results and the experimental ones, which is an indirect evidence for the good chosen model and method.

Reviewer 2 Report

The authors have presented a study in a manuscript entitled "Magnetic and optical properties of pure and ion doped MnFe2O4 nanoparticles", showing various properties of the MnFe2O4 nanoparticles compared to previous results. The following points should be considered:

1. In the abstract, could you explain the microscopic model more specifically from the method? Please also present the important results in the abstract.

2. What is the novelty and importance of this study? Both are not found in the abstract, introduction, and conclusion.

3. Please correct grammatical errors thoroughly in the whole manuscript, which is currently difficult to be understood. Using an English-language proofreading service might be helpful.

4. In general, the method and discussion are not well presented. This manuscript requires a major improvement, mainly in the method and discussion. 

5. Line 39: The section model shows several equations but the numerical method is not clearly informed. Moreover, there are several results presented in the result section, such as temperature dependence of Ms, size dependence of Curie temperature, and doping concentration dependences of TC and bandgap. How did the authors calculate those results? It is not explained in detail.

6. Line 49-54: This paragraph seems similar to that of Section 3 of Reference 24. Please check or use another form of sentence to express this paragraph.

7. Line 71: What is the purpose to explain the result of ZnFe2O4? This short paragraph does not really show a comparison with the current result. Please explain or elaborate so that the previous results of ZnFe2O4 are necessary to be presented in this paragraph.

8. Line 96-97: Mentioning the application of rare earth (RE) elements from Ref 45 seems to be unnecessary because the system in the reference is not directly related to the material studied in this manuscript.

9. Is an acknowledgment section really unnecessary for this study? Please check.

10. In the method/model, providing a crystal model or a system model might be useful for the reader to understand this study.

Overall, the reviewer suggests that this manuscript should not be accepted in this form due to the above points.

Author Response

Reviewer 2:

  1. “In the abstract, could you explain the microscopic model more specifically from the method? Please also present the important results in the abstract.”

In the Abstract we have added some explanation of the used microscopic models more specifically from the method.

The new results in the Abstract are also included.

  1. “What is the novelty and importance of this study? Both are not found in the abstract, introduction, and conclusion.”

On pp. 3-4 we have added some comments about the novelty of our work. The most theoretical papers studied the bulk MFO compounds using the density functional theory. We have for the first time considered MFO nanoparticles using microscopic models and the Green’s function theory.

  1. “Please correct grammatical errors thoroughly in the whole manuscript, which is currently difficult to be understood. Using an English-language proofreading service might be helpful.”

We have corrected the grammatical errors in the manuscript.

  1. “In general, the method and discussion are not well presented. This manuscript requires a major improvement, mainly in the method and discussion.”

The Green’s function method is that of Tserkovnikov [30], which is given now on p. 5. We go beyond the random phase approximation, taking into account all correlation functions. Moreover this method allows us to calculate also the imaginary part of the Green’s function.

The discussion is also improved.

  1. Line 39: The section model shows several equations but the numerical method is not clearly informed. Moreover, there are several results presented in the result section, such as temperature dependence of Ms, size dependence of Curie temperature, and doping concentration dependences of TC and bandgap. How did the authors calculate those results? It is not explained in detail.”

We have given additively the expressions for the spin excitation energy (3) and the electronic energies (11). It is added that \beta=1/kBT, thus it can be seen the temperature dependence of M.

The band gap energy is calculated from the s-d model, Eqs. (1), (8) and (9). Through M Eg is also temperature dependent.

We have added on pp. 6 and 7 some explanations for the exchange interaction constants on the surface and the doped states, which allow us to consider the surface, size and doping dependence of the properties. Moreover, in Eq. (1) is a factor (1-x).

  1. Line 49-54: This paragraph seems similar to that of Section 3 of Reference 24. Please check or use another form of sentence to express this paragraph.”

We have checked the two paragraphs and can remark, that the discussed properties are different, M is in other T-interval; TC increases here with decreasing NP size, whereas there it decreases; the doping dependence of TC is here considered for Co and La, whereas there for Fe and Mg; Cp has here one maximum, there – two, etc.

Let us emphasize, that our method, which cannot be changed, is to compare the radii of the doping and host ions, to see what strain appears, to choose the relation between the exchange interaction constants in the doped and undoped state and to calculate the behavior of the physical quantity in dependence on the doping concentration. We consider different doping ions in different compounds, which can lead to the same or different behavior. But by the discussion could be repetitions, which could not be removed.

  1. Line 71: What is the purpose to explain the result of ZnFe2O4? This short paragraph does not really show a comparison with the current result. Please explain or elaborate so that the previous results of ZnFe2O4 are necessary to be presented in this paragraph.”

Thank you for this remark. The referee has right. We have removed the comparison with ZnFe2O4.

  1. Line 96-97: Mentioning the application of rare earth (RE) elements from Ref 45 seems to be unnecessary because the system in the reference is not directly related to the material studied in this manuscript.”

We have removed the comment with Ref. [45].

  1. “Is an acknowledgmentsection really unnecessary for this study? Please check.”

We have added an Acknowledgement section.

  1. “In the method/model, providing a crystal model or a system model might be useful for the reader to understand this study.”

On p. 4 we have extended the presentation of the model.  

Reviewer 3 Report

In this manuscript entitled "Magnetic and optical properties of pure and ion doped MnFe2O4 nanoparticles", the authors have investigated the magnetic and optical properties of pure and ion doped MnFe2O4-bulk and nanoparticles.

The authors have found that the magnetization Ms and the Curie temperature TC increase with decreasing particle size. The band gap energy Eg decreases by Co, Mg or Ni ion substitution, but it increases by La or Gd ion doping.

The research content and method in this paper is advanced, but some issues need to be addressed before acceptance.

Some recently updated articles [Bojun Zhao, et al. Materials 16, 75 (2023)] related to the magnetic properties of strongly correlated transition metal oxides are suggested to be cited in the introduction or discussion part.

Magnetization MS (arb. units), as shown in Figure 1. Why is the unit of moment “arb. units”?

See Figure 4. How do the authors get the band gap Eg?

In Figure 5, “Applying a strong external magnetic field h the peak disappears, the transition is no longer observable (curve 3).” What does the peak represent? Moreover, the reason why the peak disappears should be explained.

Author Response

Reviewer 3:

  1. “Some recently updated articles [Bojun Zhao, et al. Materials 16, 75 (2023)] related to the magnetic properties of strongly correlated transition metal oxides are suggested to be cited in the introduction or discussion part.”

We have added the paper of Zhao et al. (2023) as Ref. [20] in the Introduction.

  1. “Magnetization MS (arb. units), as shown in Figure 1. Why is the unit of moment “arb. units”?”

The spontaneous magnetization Ms is calculated from the average of <Sz>, see Eq. (2), so that we observe an average value of Ms which is given in arbitrary units. Moreover, we make only qualitative comparisons with the experimental data for Ms.

  1. See Figure 4. How do the authors get the band gap Eg?

We have added on p. 11 the equation for the electronic energies which we need by the calculation of the band gap energy Eg.

  1. “In Figure 5, “Applying a strong external magnetic field h the peak disappears, the transition is no longer observable (curve 3).” What does the peak represent? Moreover, the reason whythe peak disappears should be explained.”

On p. 9 are given some comments for the origin of the decrease of Cp with increasing magnetic field h. The peak appears at the phase transition temperature TC where the Fe3+ ion spins undergo a long-range ferrimagnetic ordering. The decrease of the peak in Cp is probably due to the depression of spin fluctuation enhancement of the heat capacity by moments induced on the Fe atoms or is due to the magnetic quenching of localized spin fluctuations in the vicinity of the Fe atoms by increasing the magnetic fields. But it is not so easy to give a full physical explanation for this decrease, because many factors play a role in this process. 

Round 2

Reviewer 1 Report

Dear authors,

The manuscript can be accepted in the present form. All the best for your future works

Reviewer 2 Report

The authors have well addressed the points. The referee suggests that this manuscript can be accepted in this form.

Reviewer 3 Report

The paper has been improved and can be accepted.